# Periodontitis Impact in Interleukin-6 Serum Levels in Solid Organ Transplanted Patients: A Systematic Review and Meta-Analysis

**DOI:** 10.3390/diagnostics10040184

**Published:** 2020-03-27

**Authors:** Vanessa Machado, João Botelho, Joana Lopes, Mariana Patrão, Ricardo Alves, Leandro Chambrone, Gil Alcoforado, José João Mendes

**Affiliations:** 1Periodontology Department, Clinical Research Unit (CRU), Centro de Investigação Interdisciplinar Egas Moniz (CiiEM), Instituto Universitário Egas Moniz, 2829-511 Caparica, Portugal; vmachado@egasmoniz.edu.pt (V.M.); ralves@egasmoniz.edu.pt (R.A.); galcoforado@egasmoniz.edu.pt (G.A.); 2Clinical Research Unit (CRU), Centro de Investigação Interdisciplinar Egas Moniz (CiiEM), Instituto Universitário Egas Moniz, 2829-511 Caparica, Portugal; joanampmlopes@gmail.com (J.L.); mariana.rmd.patrao@gmail.com (M.P.); jmendes@egasmoniz.edu.pt (J.J.M.); 3School of Dentistry, Ibirapuera University, 04661-100 São Paulo, Brazil; leandro_chambrone@hotmail.com; 4Unit of Basic Oral Investigation (UIBO), El Bosque University, 131 A-02 Bogota, Colombia

**Keywords:** periodontitis, periodontal diseases, transplantation, interleukin-6, systematic review

## Abstract

This systematic review aimed to investigate the influence of periodontitis on post-transplant IL-6 serum levels of solid organ transplanted patients as compared to healthy subjects. Four databases (PubMed, Scholar, EMBASE, and CENTRAL) were searched up to February 2020 (PROSPERO CRD42018107817). Case-control and cohort studies on the association of IL-6 serum levels with a periodontal status of patients after solid organ transplantation were included. The risk of bias of observational studies was assessed through the Newcastle-Ottawa Scale (NOS). Random effects meta-analyses were thoroughly conducted. GRADE assessment provided quality evidence. Four case-control studies fulfilled the inclusion criteria (274 transplant recipients and 146 healthy controls), all of low risk of bias. Meta-analyses revealed significantly higher IL-6 levels in transplanted patients than healthy individuals with low-quality evidence (Mean Difference (MD): 2.55 (95% confidence interval (CI): 2.07, 3.03)). Transplanted patients with periodontitis have higher serum IL-6 levels than transplanted patients without periodontitis with moderate quality evidence (MD: 2.20 (95% CI: 1.00, 3.39)). We found low-quality evidence of higher IL-6 levels than healthy patients in patients with heart and kidney transplant. In these transplanted patients, there was moderate quality evidence that periodontitis is associated with higher IL-6 serum levels. Future research should consider the impact of such a difference in organ failure and systemic complications.

## 1. Introduction

In 2016 approximately 135,860 solid organ transplants were reported worldwide, 7.5% more than in 2015, according to the Global Observatory on Donation and Transplant (GODT) [1]. The majority were living kidney (40.2%) and liver (19.8%) transplant recipients [1] and its socioeconomic burden has been increasing [2,3]. After transplantation, chronic immunosuppression increases the risk to infections and, for this reason, its close monitoring is fundamental, since any pathology can result in systemic events and perhaps critical complications for the organ and, consequently, for the patient [4,5,6].

Periodontal diseases remain a public health problem because of their high prevalence and burden [7,8,9]. Its pathophysiologic mechanisms depict a complex interaction between the host immune system and oral bacteria, leading to inflammation and illness [10,11,12,13]. Moreover, the poor clinical control of periodontal disease causes an increase in serum inflammatory markers with systemic impact [14]. Therefore, it is of the utmost importance to study its systemic impact through inflammatory mechanisms, particularly in immunocompromised populations, like transplanted patients.

Initially, periodontal diseases were studied in transplanted immunosuppressed patients in order to investigate their behavior on an inhibited inflammatory and immunologic systems [15,16,17,18,19,20,21]. Later, several studies revealed cyclosporine-induced hyperplasia and its relations with gingival alterations [22,23,24,25,26]. Nevertheless, Harms and Bronny [27] and Meyer et al. [28] were the first to emphasize the importance of preoperative dental and periodontal treatments and the likelihood for the transplant failure. Later, numerous studies have reported poor oral health status before and after transplantation [29,30,31,32,33,34,35,36,37,38,39,40,41,42,43,44]. These reports highlighted the need for oral hygiene sensitization, as well a tight pre- and post-transplant multidisciplinary oral care [29,30,31,32,33,34,35,36,37,38,39,40,41,42,43,44]. However, literature remains contradictory regarding the impact of periodontal diseases for the organ-transplant success [31,33,34,35,43,44]. 

Cytokines and chemokines mediate the immuno-inflammatory response of periodontitis [45,46,47], are produced locally, and they regulate the amplitude and duration of the response [48]. IL-6 plays a key role in the periodontal pathogenesis pathways, and it has been consistently associated with inflammatory cell migration [48] and extended bone resorption as a result of increased RANK ligand production [47]. Additionally, IL-6 stimulates the production of inflammatory acute phase substances, like C-reactive protein (CRP), serum amyloid A, and fibrinogen [48]. 

A plethora of articles have reviewed the significant roles of IL-6 in autoimmunity [49] and allotransplantation [50,51]. Additionally, inflammation has a critical effect on the grafted organs fate [52]. When local inflammation occurs, several proinflammatory immune mediators, such as IL-6, are expressed and systemically dumped into circulation, triggering the effects on distant organ systems [53]. Uncontrolled resolutions of the inflammatory process have an enormous impact in this context [46]. Therefore, because high levels of IL-6 are greatly found in periodontitis patients, it becomes primordial to investigate the serum levels impact of periodontitis of such relevant pro-inflammatory cytokine in immunosuppressed patients. Additionally, the IL-6 levels are extremely relevant, because it might predispose transplant recipients to higher systemic inflammatory state and this should be ascertained.

With this being said, this systematic review aimed to explore the influence of periodontitis on the post-transplant IL-6 serum levels of solid organ transplanted patients, with the proposed hypothesis that transplanted patients with periodontitis have higher IL-6 serum levels than non-periodontitis transplanted patients.

## 2. Materials and Methods 

### 2.1. Protocol and Registration 

This systematic review protocol was determined, a priori, by all of the authors and was registered in the National Institute for Health Research PROSPERO, International Prospective Register of Systematic Reviews (http://www.crd.york.ac.uk/PROSPERO, ID Number: CRD42018103393). This review was structured towards the Cochrane Handbook of Systematic Reviews of Interventions [54] and reported according to the PRISMA guidelines [55] (Appendix A) and its extension for abstracts [56].

### 2.2. Focused Question

The following focused question was addressed: “Does periodontitis influence the IL-6 serum levels in organ transplanted patients?”; with the following statements: Solid Organ Transplanted Patients (Patients—P); Periodontitis (Intervention/Exposure—I); No periodontitis (Comparison—C); and, IL-6 serum levels (Outcome—O).

### 2.3. Eligibility Criteria

Both randomized clinical trials (RCTs) and observational studies (case-control and cohort studies) that investigated the periodontal condition of transplant candidates or transplanted patients were included. For the comparisons between transplant candidates and healthy patients, the studies without periodontal data before and after transplantation, without a defined control group, or without references to periodontal diagnostic criteria were excluded. Studies were excluded if not reporting the exclusion of patients that might directly affect the systemic inflammatory status (e.g., rheumatoid or autoimmune diseases), smokers, had history of acute infections and antibiotic use during the previous four months, and had periodontal treatment within the previous year. 

### 2.4. Information Sources and Search

PubMed, Google Scholar, and CENTRAL (The Cochrane Central Register of Controlled Trials) were searched up to and including November 2019 to expedite the detection of potentially eligible studies in this SR. There were no limitations regarding the year of publication or language. The reference lists of included articles and relevant reviews were manually searched. Gray literature was searched through OpenGray (http://www.opengrey.eu). The authors were contacted when necessary for additional data or clarifications.

We merged keywords and subject headings in accordance with the thesaurus of each database and applied exploded subject headings. Our PubMed search strategy was based on the following algorithm: ((chronic periodontitis OR periodontitis, chronic OR adult periodontitis OR periodontitis, adult OR periodontal disease OR alveolar bone loss OR attachment loss, periodontal OR periodontal pocket) and (transplantation OR organ transplantation OR transplantation, organ OR tissue transplantation OR transplantation, tissue OR heart transplantation OR kidney transplantation OR liver transplantation OR lung transplantation OR pancreas transplantation) AND (interleukin-6 OR IL-6 OR interleukin OR chemokines OR cytokines)). Our Google Scholar search syntax was: “Periodontitis“%” Oral Health“%” Transplantation“%” Interleukin-6”. The manual search was conducted on the following journals: Journal of Clinical Periodontology, Periodontology 2000, Journal of Periodontology, and Journal of Periodontal Research.

### 2.5. Study Selection

Study selection was initially conducted by two trained authors (VM and JB), who screened the titles and/or abstracts of retrieved studies. Following full-text assessment, the interexaminer reliability was calculated via kappa statistics. Final selection of studies was performed by two authors independently (VM and JB) by reviewing the full text based on the inclusion criteria above. Any disagreements were resolved by discussion. 

### 2.6. Data Extraction Process and Data Items

A predefined table was used to conduct data extraction. The extracted data included: the first author’s name, study design, publication year, country of origin of the research, number of cases and participants, gender, transplanted organ/tissue, post-transplantation years, mean age in years, periodontal diagnostic criteria, IL-6 serum levels in pg/mL, type of IL-6 quantification laboratory method, diabetes, smokers, and clinical measures. Clinical measures included percentage with periodontitis, probing depth (PD), plaque index (PI), missing teeth, bleeding on probing (BOP), and clinical attachment loss (CAL). Two reviewers independently extracted all of the data (VM and JB) with a consensus on all the aspects. 

### 2.7. Risk of Bias in Individual Studies 

Case-control and cohort studies were appraised through the Newcastle-Ottawa (NOS) Scale. “Stars” (points) were attributed for each methodologic quality standard, and each study could attain a maximum of eight points. Studies with seven to eight points (80% or more of the domains satisfactorily fulfilled) were arbitrarily considered to be of high quality, studies with five to 6 stars were of medium quality, and studies with less than five stars were of low methodologic quality. Discussion resolved the disagreements between the review authors (VM and JB) over the risk of bias in any particular studies, with the involvement of a third review author where necessary (LC). 

### 2.8. GRADE Assessment of Overall Evidence Quality

Two reviewers (V.M. and J.B.) appraised the strength and certainty of the body of evidence for the outcomes through the GRADE approach [57,58]. The GRADE assessment was performed for “IL-6 serum of transplanted patients vs healthy patients” and “IL-6 levels of transplanted patients with periodontitis vs no periodontitis” using the online tool https://gradepro.org/.

According to GRADE terminology [58], the quality of the overall body of evidence for each outcome was: “high certainty” for high-quality evidence; “moderate certainty” for moderate-quality evidence; “may/may not” for low-quality evidence; and, “large uncertainty” for very low-quality evidence.

### 2.9. Summary Measures and Synthesis of Results 

The median and interquartile range reported in selected studies for IL-6 serum levels for cases and controls were converted to mean and standard deviations following Hozo, Djulbegovic, and Hozo procedure [59], under the assumption of normal distribution. We performed two different sets of meta-analyses. Primarily, we evaluated whether the IL-6 serum levels of transplanted patients were different from healthy patients. In the second analysis, the serum levels of IL-6 of transplanted patients with periodontitis were compared to periodontally health transplanted patients or severe periodontitis with mild or moderate periodontitis.

DerSimonian-Laird random-effects model [60] was performed, as previously described [61] while using R version 3.4.1 (R Studio Team 2018). Forest plots were rendered to visualise estimates and the correspondent 95% Confidence Intervals (CIs). All of the random-effects meta-analysis and forest plots were performed using ‘meta’ package [61]. The quantity I2 assessed the degree of dispersion of effect sizes (ES) estimates and the overall homogeneity was calculated through the Chi-square (χ^2^) test [62]. Meta-analyses were carried out for each type of sample. All of the tests were two-tailed with alpha being set at 0.05, except for the homogeneity test, whose significance level cutoff was considered to be 0.10 due to the low power of the χ^2^ test with a limited amount of studies. Publication bias analysis was planned to be performed if, at least, we had 10 or more studies included [54]. Overall ES estimates were reported with 95% confidence intervals (CI).

## 3. Results

### 3.1. Study Selection 

A total of 445 records were identified and judged against the eligibility criteria after electronic searches and duplicates removal. Titles and abstracts review resulted in 437 exclusions. Subsequently, the remaining eight full publications were reviewed, leaving a final number of four case-control studies to be included in this SR (Figure 1 and Appendix A). After full-text screening, interexaminer reliability was considered to be excellent (kappa score = 0.973).

### 3.2. Study Characteristics

Table 1 lists the characteristics of the included studies. Four case-control studies were included from the United States [63,64], Poland [32], and Turkey [65]. The study sample sizes ranged from 40 [65] to 162 participants [64]. Globally, this review comprised a total of 420 participants (237 males and 183 females), 274 transplant recipients (167 males and 107 females), and 146 healthy controls (70 males and 76 females). 

### 3.3. Risk of Bias within Studies

Table 2 (NOS Scale scores) shows the risk of bias assessment for the included studies. All four studies were considered of low risk of bias and, therefore, of high quality [32,63,64,65]. Specifically, one study described an alternative periodontal case definition [64].

### 3.4. Synthesis of Results

#### 3.4.1. IL-6 Levels of Transplanted Patients vs. Healthy Patients

Data from four studies [32,63,64,65], including 420 patients reported IL-6 serum and GCF levels of transplanted and healthy patients. All 274 transplant recipients were heart and kidney transplanted patients. The results of the meta-analysis shows a moderately increased value of serum IL-6 levels in the transplanted patients when compared to healthy controls, with an average of 2.55 pg/mL above (*p* < 0.01). The pooled mean difference (95% confidence interval (CI)) between the test and control group was 2.55 (2.07–3.03). χ^2^ test revealed moderate heterogeneity, with I^2^ = 57% (Figure 2). The overall evidence quality was graded as low quality of evidence (Appendix A).

#### 3.4.2. IL-6 Levels of Transplanted Patients with Periodontitis vs. no Periodontitis

Likewise, three studies [32,63,64], including 254 patients, reported IL-6 serum levels of transplanted with and without periodontitis. The overall result suggests a slightly increased value of serum IL-6 levels in transplanted patients with periodontitis when compared to transplant recipients without periodontitis, with an average of 2.20 pg/mL above (*p* < 0.01) (MD (95% CI): 2.20 (1.00–3.39)) (Figure 3). The χ^2^ test revealed excellent heterogeneity, with I^2^ = 21% (Figure 3). The overall evidence quality was graded as moderate quality of evidence (Appendix A).

## 4. Discussion

### 4.1. Summary of Main Findings

This systematic reviewed aimed to investigate the influence of periodontitis on post-transplant IL-6 serum levels transplant recipients and, therefore, if the patient is predisposed to a superior systemic inflammatory state. We began by investigating whether the IL-6 levels from transplanted patients would differ from healthy people. The meta-analysis reported an average of 2.55 pg/mL increased serum levels of this inflammatory cytokine when compared to a healthy systemic state, and previous research confirms this result [46,49,50,51,66,67]. This result was of low-quality of evidence due to the moderated inconsistency and studies included few patients with wide confidence interval around the estimate of the effect.

Following, the influence of periodontitis on post-transplant IL-6 serum levels was sourced from three systematically selected case-control studies. Comprehensively, periodontitis predisposes transplant recipients to higher systemic inflammatory state, with an average of 2.20 pg/mL above the IL-6 levels. This result was of moderate quality evidence, mainly because studies with few patients and a wide confidence interval around the estimate of the effect.

These results might be impactful to the current available evidence. 1) This is the first systematic review to provide concrete evidence on the inflammatory burden of periodontitis on transplant recipients. 2) The results point to a potential impact of periodontitis, although the possible systemic complications shall be investigated. 3) Both dental practitioners and transplant staff should be aware of this association and they should consider periodontal health as a critical factor for the patient.

### 4.2. Quality of the Evidence, Limitations and Potential Biases in the Review Process

The strengths of this systematic review are the strict and predetermined protocol, and rigorous methodology implemented in each phase. In addition, all of the studies were carried out in hospital settings and they are fully representative transplanted populations, which allows for generalized conclusions. Besides, the quality of evidence of the results were appraised through the GRADE approach.

However, the selected studies had too heterogeneous periodontal diagnostic criteria, since none used the same diagnostic protocol. Shaqman et al. [64] used the CDC/AAP definition criteria, Blach et al. [32] used CPTIN scores, Gürkan et al. [65] used the 1999 AAP, and Ioannidou et al. [64] used an alternative case definition. Further, Blach et al. [32] included moderate periodontitis cases with healthy patients in the control group, which certainly contributed to a greater heterogeneity to the results. Hence, a consensus on the appropriate periodontitis case definition to use is warranted in future studies.

Regarding the type of transplant recipients included in this study, only studies with kidney and heart transplants were available. While renal transplantation accounts for the most prevalent type, liver transplants are not represented in this study, despite being prevalent worldwide (WHO 2016). Hereafter, the behavior of the inflammatory markers in more different types of transplant organ receives is warranted.

The likelihood of negative outcomes for the transplant remains unclear, although some studies concluded that stem-cells transplant recipients with periodontal diseases have a higher risk of developing infectious complications [31,68]. Additionally, inflammation and inflammatory injury, adaptive immunity and metastatic infections are the three major mechanisms responsible for periodontitis’ systemic impact [48]. Besides, oral pathogens can survive in the bloodstream and they adhere at non-oral body sites being responsible for infections, such as endocarditis, lung infections, and liver and brain abscesses [48].

Further, increased serum levels of IL-6 contribute greatly to periodontal destruction and this higher prevalence is of concern for the progression of periodontitis [69,70]. The included studies do not have available data on organ function and the survival rates for the studied cohorts, although a bidirectional association cannot be disregarded. Therefore, if such difference is enough to result in organ failure and systemic complications, this is a matter that should be addressed in the future.

Moreover, of the included studies, two had most of the sample being composed of diabetic patients (63,64), while, in the other study, diabetic patients were poorly represented [32]. Elevated IL-6 levels are significantly associated with increased risk of type 2 diabetes [71], and, when considering the post-transplant mean age of studies, this could contribute to the high percentage observed, although this is not observed in Blach et al. [32].

Another relevant limitation might be the IL-6 quantification technique used, when considering that most studies [63,64,65] used ELISA quantification and Blach et al. [32] measured through immunoradiometric method (IRMA). Although with some contradiction, the literature seems to show significant differences between these two methodologies in serum readings [72,73,74,75]. Moreover, the moderate heterogeneity of the results on IL-6 levels between transplanted and healthy patients might be explained by the fact that it resulted from studies with serum and gingival crevicular fluid.

Furthermore, several systematic reviews demonstrated that periodontal treatment reduces the inflammation biomarkers of atherosclerotic disease and diabetic patients, including IL-6 [76,77]. When considering the latter, it is likely that periodontal treatment will have positive results on transplant candidates or transplant patients and, therefore, every patient diagnosed with periodontitis should be immediately treated. Certainly, periodontal health might putatively contribute to overall success of the transplantation, complications, healing, and future health with immunosuppression. Notwithstanding, future studies should focus on the effect of periodontal treatment on transplant candidates and transplant recipients. 

## 5. Conclusions

In patients with heart and kidney transplant, there is low-quality evidence of higher IL-6 levels than healthy patients. There is moderate quality evidence that the IL-6 serum levels are increased in the transplanted individuals with periodontitis in comparison with cases without periodontitis. Future research should consider the impact of such a difference in organ failure and systemic complications.

## Figures and Tables

**Figure 1 diagnostics-10-00184-f001:**
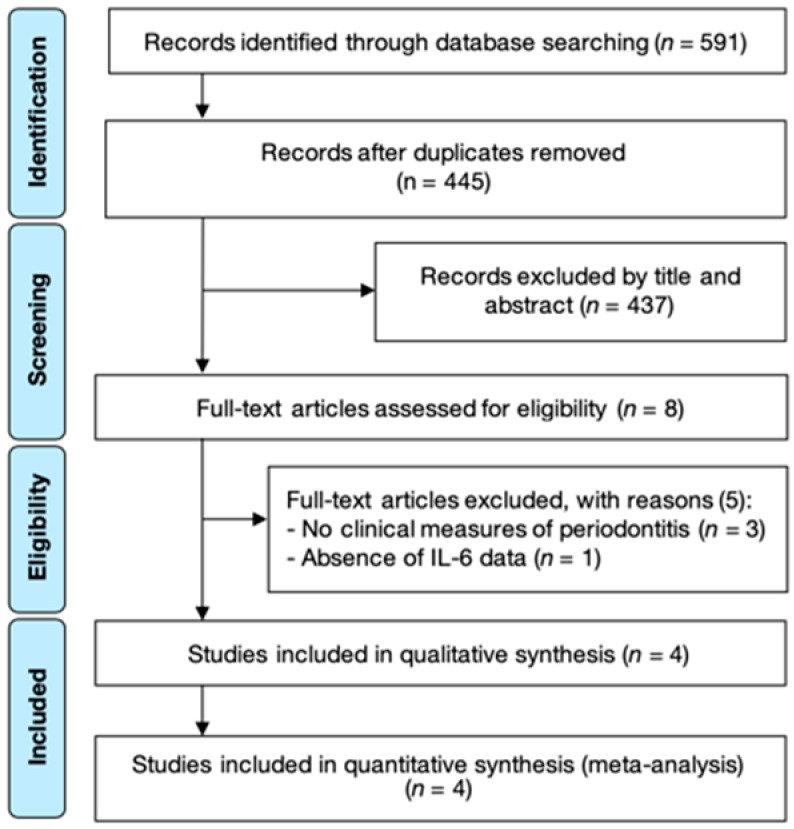
PRISMA flow-chart representing the results of the workflow to identify eligible studies.

**Figure 2 diagnostics-10-00184-f002:**
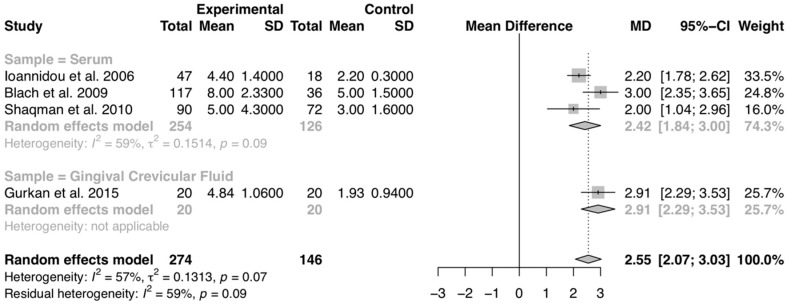
Forest plot of studies with serum IL-6 mean values comparing transplanted and healthy patients. Mean effect size estimates have been calculated with 95% confidence intervals and are shown in the figure. Area of squares represents sample size, continuous horizontal lines and diamonds width represents 95% confidence interval. The bottom diamond center and the vertical dotted line represent the overall pooled estimate.

**Figure 3 diagnostics-10-00184-f003:**
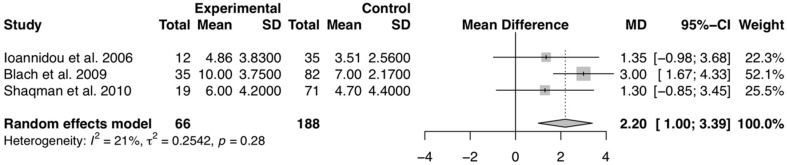
Forest plot of studies with serum IL-6 mean values comparing transplanted patients with and without periodontitis. Mean effect size estimates have been calculated with 95% confidence intervals and are shown in the figure. Area of squares represents sample size, continuous horizontal lines and diamonds width represents 95% confidence interval. The bottom diamond center and the vertical dotted line represent the overall pooled estimate.

**Table 1 diagnostics-10-00184-t001:** Characteristics of the included studies.

Reference	Healthy Patients (Female/Male)	Transplanted Patients (Female/Male)	Test vs. Control	Transplanted Organ(s)	Post-Transplantation Years (Mean ± SD)	Age (Mean ± SD)	Diagnostic Criteria	IL-6 Serum Levels (Mean ± SD) (Control/Test) (pg/mL)	IL-6 Quantification Laboratory Method	Transplanted Patients with Diabetes (%)	Smokers (%)	Mean PD (Mean ± SD) (mm)	Mean CAL (Mean ± SD) (mm)	Mean BoP (Mean ± SD) (%)	Mean PS (Mean ± SD) (%)	Mean Missing Teeth (Mean ± SD) (n)
Ioannidou et al., 2006 (USA) (Connecticut) [63]	18 (9/9)	47 (20/27)	12 with CP vs. 35 with no CP	Kidney and Heart	5.8 ± 4.3	55.0 ± 9.7	At least one site with > 5 mm PD requiring scaling and root planing under local anesthesia	4.86 ± 3.83 / 3.51 ± 2.56	ELISA (Diagnostic Products, Los Angeles, USA)	24 (51)	21 (44.9)	2.7 ± 0.5	2.9 ± 0.5	14.7 ± 13.8	38.9 ± 29.9	7.4 ± 4.9
Blach et al., 2009 (Poland) (Katowice) [32]	36 (13/23)	117 (40/77)	35 with severe CP vs. 82 with no or moderate CP	Kidney	3.4 ± 0.7	42.5 ± 3.1	CPITN	10.00 ± 3.75 / 7.00 ± 2.17	IRMA (Biosource Europe S.A., Belgium)	6 (5.1)	21 (17.7)	NS	NS	NS	NS	NS
Shaqman et al., 2010 (USA) (Connecticut) [64]	72 (43/29)	90 (38/52)	19 with severe CP vs. 71 with no CP	Kidney and Heart	6.9 ± 4.8	53.0 ± 12.0	Presence of ≥2 interproximal sites with CAL ≥6 mm and one interproximal site with a PD ≥5 mm (CDC/AAP definition criteria)	6.00 ± 4.20 / 4.70 ± 4.40	ELISA (Diagnostic Products, Los Angeles, USA)	50 (56.0)	0 (0)	2.7 ± 0.5	2.9 ± 0.7	17.0 ± 16.7	46.6 ± 30.0	4.2 ± 4.9
Gürkan et al. (2015) (Turkey) (Izmir) [65]	11/9	9/11	20 transplant vs. 20 healthy	Kidney	4.0 ± 2.7	42.0 ± 10.3	AAP (Armitage 1999)	4.84 ± 1.06 / 1.93 ± 0.94	ELISA	NS	NS	2.44 ± 0.6	NS	NS	2.42 ± 0.5	NS

BMI—Body Mass Index, BoP—Bleeding on Probing, CAL—Clinical Attachment Loss, CDC/AAP—Centers for Disease Control/American Academy of Periodontology, CP—Chronic Periodontitis, CPITN—Community Periodontal Index of Treatment Needs, ELISA—Enzyme-Linked Immunosorbent Assay, IL-6—Interleukin-6, IRMA—immunoradiometric, NS—Not stated, PD—Probing Depth, PS—Plaque Score.

**Table 2 diagnostics-10-00184-t002:** Newcastle-Ottawa Scale (NOS) for case-control studies in the systematic review according to the eight-items.

Article	Selection				Comparability	Outcome			Score (Risk of Bias)
	1	2	3	4	5	6	7	8	
Ioannidou et al. (2006) [63]	c	a	a	a	a	a	a	a	7 (low)
Blach et al. (2009) [32]	a	a	a	a	a	a	a	a	8 (low)
Shaqman et al. (2010) [64]	a	a	a	a	a	a	a	a	8 (low)
Gürkan et al. (2015) [65]	c	a	a	a	a	a	a	a	7 (low)

1—Is the case definition adequate?; 2—Representativeness of the cases; 3—Selection of controls; 4—Definition of controls; 5—Comparability of cohorts on the basis of the design or analysis; 6—Ascertainment of outcome; 7—Save method of ascertainment for cases and controls; 8—Non-response rate.

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
