# Peer review of "Periodontitis Impact in Interleukin-6 Serum Levels in Solid Organ Transplanted Patients: A Systematic Review and Meta-Analysis"

_diagnostics, 2020, doi:10.3390/diagnostics10040184_

Round 1

Reviewer 1 Report

The study "Periodontitis Impact in Interleukin-6 Serum Levels in Solid Organ Transplanted Patients: A Systematic Review and Meta-Analysis" (diagnostics-748767) aims to review the existing literature regarding a possible impact of periodontitis in patients subjected to a heart or kidney transplantation.

The manuscript is well designed, the review systematics correct and according to the actual and relevant guidelines, and the effort for the study is highly appreciated.

Some issues however need further clarification and some additional information is needed. Then, some corrections should be performed in order to enhance the paper's quality.

  • English language and grammar: The wole text should be revised carefully in order to eliminate typos (e.g. "February 202020, ln19), grammar mistakes (e.g. "...worrying...grounding...", ln 54ff, "DOES" instead of "Do periodontitios...."ln84) and to replace odd terms (e.g. "crucial" or "critical" instead of "catastrophic", ln 49,  "uncontrol" ln 44,  "triggering" instead of "nettling", ln 68).

Where is Table 4 (ln210)?

Ln70ff. A hypothesis might be defined - and lateron confirmed or rejected in order to better structure the manuscript. Something that remains somehow unclear is: If periodontitis rises the level of IL-6 (what is important for patients with organ transplants) what exactly is the rational to assess whether in transplanted patients the IL-6 level (most probably suppressed after organ transplant) is higher in presence of periodontitis? (Sure, there IS a good rational, but it should be pointed out much more clearly at the end of the introduction).

Fig.2 is in black-and-white. Legend explains coloured elements. (Should be corrected). In this forest blot data of IL-6 levels from different media get merged, i.e. gingival fluid and serum. This apprach is highly discussable and should either be revised or highlighted and discussed properly in the respective discussion section.

Conclusions: The specification with/without periodontitis must not be forgotten.

"This polymicrobial disruption portray..."ln 43 - the term pm disruption is not well-chosen in the present context. And it is singular, so it would be portrays, but "depicts" might be the better term.

"Repercussion"ln54, "impact" might be the better term

ln46ff - some interaction of periodontitis and the regarding diseases is discussed critically in literature, but the statement is too strong in its current form.

"periodontal" ln51. Since the symptom is gingiva hyperplasty the correct term is "gingival"

ln59 better divide statement in (at least) two sentences in order to render meaning clearer

"Numberless"ln65 actually there IS a number for these studies, but there are many of them. "Plethora" might be the fitting term.

Author Response

Dear Reviewer 1,

We are pleased with the opportunity to revise and resubmit our manuscript “Impact in Interleukin-6 Serum Levels in Solid Organ Transplanted Patients: A Systematic Review and Meta-Analysis” (diagnostics-748767).

Manuscript changes are highlighted in the revised manuscript. Our point-by-point responses to all comments are outlined and detailed below. We hope that you find our responses satisfying.

We hope the revised manuscript will better suit Diagnostics. We are happy to consider further revisions and we thank you for your continued interest in our research.

The study "Periodontitis Impact in Interleukin-6 Serum Levels in Solid Organ Transplanted Patients: A Systematic Review and Meta-Analysis" (diagnostics-748767) aims to review the existing literature regarding a possible impact of periodontitis in patients subjected to a heart or kidney transplantation.

The manuscript is well designed, the review systematics correct and according to the actual and relevant guidelines, and the effort for the study is highly appreciated.

Some issues however need further clarification and some additional information is needed. Then, some corrections should be performed in order to enhance the paper's quality.

English language and grammar: The wole text should be revised carefully in order to eliminate typos (e.g. "February 202020, ln19), grammar mistakes (e.g. "...worrying...grounding...", ln 54ff, "DOES" instead of "Do periodontitios...."ln84) and to replace odd terms (e.g. "crucial" or "critical" instead of "catastrophic", ln 49,  "uncontrol" ln 44,  "triggering" instead of "nettling", ln 68).

Answer: We appreciate these remarks, and all were resolved accordingly.

Where is Table 4 (ln210)?

Answer: We apologize for this lapse, we meant to state Appendix B. This was corrected (Page 8, Line 282).

Ln70ff. A hypothesis might be defined - and lateron confirmed or rejected in order to better structure the manuscript. Something that remains somehow unclear is: If periodontitis rises the level of IL-6 (what is important for patients with organ transplants) what exactly is the rational to assess whether in transplanted patients the IL-6 level (most probably suppressed after organ transplant) is higher in presence of periodontitis? (Sure, there IS a good rational, but it should be pointed out much more clearly at the end of the introduction).

Answer: We have proposed a hypothesis as suggested “, with the proposed hypothesis that transplanted patients with periodontitis have higher IL-6 serum levels than non-periodontitis transplanted patients.” (Page 2, Lines 83-84). Also, we clarified the rationale for investigating IL-6 levels by stating “Also, IL-6 levels are extremely relevant because it may predispose transplant recipients to higher systemic inflammatory state and this should be ascertained.” (Page 2, Lines 78-80).

Fig.2 is in black-and-white. Legend explains coloured elements. (Should be corrected). In this forest blot data of IL-6 levels from different media get merged, i.e. gingival fluid and serum. This apprach is highly discussable and should either be revised or highlighted and discussed properly in the respective discussion section.

Answer: We corrected the legends appropriately by removing all color references. Moreover, we discussed the important point you raised by stating “Moreover, the moderate heterogeneity of the results on IL-6 levels between transplanted and healthy patients might be explained by the fact that it resulted from studies with serum and gingival crevicular fluid.” (Page 10, Lines 325-327).

Conclusions: The specification with/without periodontitis must not be forgotten.

Answer: We have rephrased to “In patients with heart and kidney transplant, there is low-quality evidence of higher IL-6 levels than healthy patients. There is moderate quality evidence that IL-6 serum levels are increased in transplanted individuals with periodontitis in comparison with cases without periodontitis.” (lines 337-339).

"This polymicrobial disruption portray..."ln 43 - the term pm disruption is not well-chosen in the present context. And it is singular, so it would be portrays, but "depicts" might be the better term.

We: we have rephrased the sentence: “Its pathophysiologic mechanisms depict” (Page 1, Line 43).

"Repercussion"ln54, "impact" might be the better term

Answer: we have changed for “impact” (Page 2, line 52).

ln46ff - some interaction of periodontitis and the regarding diseases is discussed critically in literature, but the statement is too strong in its current form.

Answer: we agree with the reviewer. Therefore we rephrased the sentence for “causes an increase in serum inflammatory markers with systemic impact [14]” (Page 2, Line 52).

"periodontal" ln51. Since the symptom is gingiva hyperplasty the correct term is "gingival"

ln59 better divide statement in (at least) two sentences in order to render meaning clearer

Answer: We have changed the term “periodontal” to “gingival” (page 2, Line 57) and rephrased the statement in two sentences: “Later, numerous studies have reported poor oral health status before and after transplantation [32, 33, 42–47, 34–41]. These reports highlighted the need for oral hygiene sensitization, as well a tight pre- and post-transplant multidisciplinary oral care [32, 33, 42–47, 34–41].” (Page 2, Lines 60-62).

"Numberless"ln65 actually there IS a number for these studies, but there are many of them. "Plethora" might be the fitting term.

Answer: we have changed “Numberless” for “A plethora of” (Page 2, Line 71).

Reviewer 2 Report

Thank you for submitting your manuscript entitled "Periodontitis Impact in Interleukin-6 Serum Levels in Solid Organ Transplanted Patients: A Systematic Review and Meta-Analysis" to Diagnostics. I am sure that you spent considerable time and effort on performing this study.

In the abstract, page 1 line 18, there is mistake, “…were searched up to February 202020…”. And please, write the number of the patients included (transplant recipients and controls).

Maybe it Will be important to write that a consensus about using same definition of a case with periodontitis will be necesary, since none used the same diagnostic protocol, four manuscripts included and four definitions.

Only a question, Is there in the flowchart with any mistakes?, since in the flowchart is written that there are 3 included manuscripts in the SR and 3 in the MA.

This manuscript meet PRISMA standards, NOS scale, GRADE…really well done. It is well written paper.

Best wishes

Author Response

Dear Reviewer 2,

We are pleased with the opportunity to revise and resubmit our manuscript “Impact in Interleukin-6 Serum Levels in Solid Organ Transplanted Patients: A Systematic Review and Meta-Analysis” (diagnostics-748767).

Manuscript changes are highlighted in the revised manuscript. Our point-by-point responses to all comments are outlined and detailed below. We hope that you find our responses satisfying.

We hope the revised manuscript will better suit Diagnostics. We are happy to consider further revisions and we thank you for your continued interest in our research.

Thank you for submitting your manuscript entitled "Periodontitis Impact in Interleukin-6 Serum Levels in Solid Organ Transplanted Patients: A Systematic Review and Meta-Analysis" to Diagnostics. I am sure that you spent considerable time and effort on performing this study.

In the abstract, page 1 line 18, there is mistake, “…were searched up to February 202020…”. And please, write the number of the patients included (transplant recipients and controls).

Answer: We appreciate these remarks, and all were resolved accordingly. Also, we added “(274 transplant recipients and 146 healthy controls)” (Page 1, Lines 23-24).

Maybe it Will be important to write that a consensus about using same definition of a case with periodontitis will be necesary, since none used the same diagnostic protocol, four manuscripts included and four definitions.

Answer: Within this remark in consideration, we stated “Hence, a consensus on the appropriate periodontitis case definition to use is warranted in future studies.” (Page 10, Lines 298-299).

Only a question, Is there in the flowchart with any mistakes?, since in the flowchart is written that there are 3 included manuscripts in the SR and 3 in the MA.

Answer: the reviewer is absolutely right. We have corrected the flowchart accordingly (Page4, Line 204).

This manuscript meet PRISMA standards, NOS scale, GRADE…really well done. It is well written paper.

Best wishes

Answer: We appreciate these very kind words and the time spent reviewing our manuscript.

Reviewer 3 Report

In Data extraction process and data items paragraph

In periodontal diagnostic criteria, why didn't you consider the radiological criteria in clinical measures?

Author contributions 

The individual contributions must be provided

Author Response

Dear Reviewer 3,

We are pleased with the opportunity to revise and resubmit our manuscript “Impact in Interleukin-6 Serum Levels in Solid Organ Transplanted Patients: A Systematic Review and Meta-Analysis” (diagnostics-748767).

Manuscript changes are highlighted in the revised manuscript. Our point-by-point responses to all comments are outlined and detailed below. We hope that you find our responses satisfying.

We hope the revised manuscript will better suit Diagnostics. We are happy to consider further revisions and we thank you for your continued interest in our research.

In Data extraction process and data items paragraph

In periodontal diagnostic criteria, why didn't you consider the radiological criteria in clinical measures?

Answer: In fact, we did consider the radiological criteria in clinical measures, however there were no studies following the remaining inclusion criteria at the same time.

Author contributions 

The individual contributions must be provided

Answer: we completed the Individual contributions section accordingly: “: Conceptualization, Methodology, V.M. and J.B.; formal analysis, J.B.; investigation, V.M. and J.B.; data curation, V.M. and J.B.; supervision, L.C.; project administration, J.J.M. All authors have written, reviewed, edited, read and agreed to the published version of the manuscript.” (Page 11, Lines 346-348).

Round 2

Reviewer 1 Report

Congrats for the manuscript!

Reviewer 3 Report

The research article appears to be sound and well conducted. 

Well understanding that the periodontitis diagnostic criteria and classification changed a lot over time and so it is hard to compare the different paper, it could be useful to compare and discuss the classification used and the results obtained in the 4 paper selected.